# Biological Activity and Antibiofilm Molecular Profile of *Citrus aurantium* Essential Oil and Its Application in a Food Model

**DOI:** 10.3390/molecules25173956

**Published:** 2020-08-30

**Authors:** Miroslava Kačániová, Margarita Terentjeva, Lucia Galovičová, Eva Ivanišová, Jana Štefániková, Veronika Valková, Petra Borotová, Przemysław Łukasz Kowalczewski, Simona Kunová, Soňa Felšöciová, Eva Tvrdá, Jana Žiarovská, Renáta Benda Prokeinová, Nenad Vukovic

**Affiliations:** 1Department of Fruit Science, Viticulture and Enology, Faculty of Horticulture and Landscape Engineering, Slovak University of Agriculture, Tr. A. Hlinku 2, 94976 Nitra, Slovakia; miroslava.kacaniova@gmail.com (M.K.); l.galovicova95@gmail.com (L.G.); veronika.valkova@uniag.sk (V.V.); 2Department of Bioenergetics, Food Analysis and Microbiology, Institute of Food Technology and Nutrition, University of Rzeszow, Cwiklinskiej 1, 35-601 Rzeszow, Poland; 3Institute of Food and Environmental Hygiene, Faculty of Veterinary Medicine, Latvia University of Life Sciences and Technologies, K. Helmaņaiela 8, LV-3004 Jelgava, Latvia; margarita.terentjeva@llu.lv; 4Department of Technology and Quality of Plant Products, Faculty of Biotechnology and Food Sciences, Slovak University of Agriculture, Tr. A. Hlinku 2, 94976 Nitra, Slovakia; eva.ivanisova@uniag.sk; 5AgroBioTech Research Centre, Slovak University of Agriculture, Tr. A. Hlinku 2, 94976 Nitra, Slovakia; jana.stefanikova@uniag.sk (J.Š.); petra.borotova@uniag.sk (P.B.); 6Institute of Food Technology of Plant Origin, Poznań University of Life Sciences, 31 Wojska Polskiego St., 60-624 Poznań, Poland; przemyslaw.kowalczewski@up.poznan.pl; 7Department of Food Hygiene and Safety, Faculty of Biotechnology and Food Sciences, Slovak University of Agriculture, Tr. A. Hlinku 2, 949 76 Nitra, Slovakia; simona.kunova@uniag.sk; 8Department of Microbiology, Faculty of Biotechnology and Food Sciences, Slovak University of Agriculture, Tr. A. Hlinku 2, 949 76 Nitra, Slovakia; sona.felsociova@uniag.sk; 9Department of Animal Physiology, Faculty of Biotechnology and Food Sciences, Slovak University of Agriculture, Tr. A. Hlinku 2, 949 76 Nitra, Slovakia; eva.tvrda@uniag.sk; 10Department of Plant Genetics and Breeding, Faculty of Agrobiology and Food Resources, Slovak University of Agriculture, Tr. A. Hlinku 2, 949 76 Nitra, Slovakia; jana.ziarovska@uniag.sk; 11Department of Statistics and Operations Research, Faculty of Economic and Management, Slovak University of Agriculture, Tr. A. Hlinku 2, 949 76 Nitra, Slovakia; renataprokein@gmail.com; 12Department of Chemistry, Faculty of Science, University of Kragujevac, P.O. Box 12, 34000 Kragujevac, Serbia

**Keywords:** bitter orange essential oil, biological activity, *Stenotrophomonas maltophilia*, *Bacillus subtilis*, *Penicillinum*, glass, wood, food

## Abstract

The main aim of the study was to investigate the chemical composition, antioxidant, antimicrobial, and antibiofilm activity of *Citrus aurantium* essential oil (CAEO). The biofilm profile of *Stenotrophonomonas maltophilia* and *Bacillus subtilis* were assessed using the mass spectrometry MALDI-TOF MS Biotyper and the antibiofilm activity of *Citrus aurantium* (CAEO) was studied on wood and glass surfaces. A semi-quantitative composition using a modified version was applied for the CAEO characterization. The antioxidant activity of CAEO was determined using the DPPH method. The antimicrobial activity was analyzed by disc diffusion for two biofilm producing bacteria, while the vapor phase was used for three penicillia. The antibiofilm activity was observed with the agar microdilution method. The molecular differences of biofilm formation on different days were analyzed, and the genetic similarity was studied with dendrograms constructed from MSP spectra to illustrate the grouping profiles of *S. maltophilia* and *B. subtilis*. A differentiated branch was obtained for early growth variants of *S. maltophilia* for planktonic cells and all experimental groups. The time span can be reported for the grouping pattern of *B. subtilis* preferentially when comparing to the media matrix, but without clear differences among variants. Furthermore, the minimum inhibitory doses of the CAEO were investigated against microscopic fungi. The results showed that CAEO was most active against *Penicillium crustosum*, in the vapor phase, on bread and carrot in situ.

## 1. Introduction

*Citrus aurantium* L. (*Rutaceae*) or bitter orange is not only used in the food industry, but it is also well-known for its application in the treatment of anxiety [1], lung, and prostate cancers [2], gastrointestinal diseases and obesity [3,4]. *C. aurantium* acts as an appetite suppressant and was prioritized for the replacement of ephedra-containing (*Ephedra sinica* L.) weight loss products [4,5,6]. Fruits of *C. aurantium* are a source of flavonoid-type compounds with diverse biological effects [7,8,9]. Additionally, the presence of flavonoid glycosides [10], biogenic amines, and flavanones has been reported [11,12]. Previous studies were focused on antimicrobial and health-promoting activities of different parts of *C. aurantium*, such as fruits and flowers, as well as essential oils obtained thereof [13,14,15]. Furthermore, substantial research has been focused on the chemical composition of essential oils prepared from various parts of *Citrus aurantium* growing in Pakistan [16], Brazil [17,18], Morocco [19], Iran [20,21], Croatia [22], northern Tunisia [23], India [24], and Algeria [25,26]. Essential oils of *C. aurantium* were reported to be a source of bioactive compounds with antimicrobial [19,21,23,27], antioxidant [19,22,23], anti-inflammatory [18], and anti-anxiety properties [20].

The ability of bacteria to adhere to the surface provides them with a range of physiological benefits and advantages for their survival, including biofilm formation [28]. *Stenotrophomonas maltophilia* is a gram-negative opportunistic nosocomial pathogen [29,30], although its role in the pathogenesis of infection in immunocompromised and hospitalized patients often remains unclear [31,32]. Studies on the virulence factors of *S. maltophila* are scarce [33,34]. *S. maltophilia* was reported to form biofilms on abiotic surfaces [35,36,37], with significant differences among clinical isolates of the bacterium. *Bacillus subtilis* is a gram-positive, spore-forming bacterium that is ubiquitous in the soil and rhizosphere environment. It is a well-recognized biofilm-producer on plant root systems, ensuring the protection of plants against numerous pathogens. Furthermore, the presence of *B. subtilis* biofilms has been linked to plant growth promotion [38,39]. The genus *Penicillium* are fundamentally saprophytic and ubiquitous fungi, which are characterized as producers of novel bioactive compounds, blockbuster drugs, such as penicillin [40] and the anticholesterolemic agent compactin [41], miscellaneous antitumor products [42], and mycotoxins contaminating food [43].

Natural plant materials have been used as food preservatives against both bacteria and fungi to control organoleptic changes, off-flavors, and mycotoxin production [44,45,46]. Natural mixtures have been applied to a variety of foodstuff, including bakery products and vegetables; however, few data on the antifungal activity of natural plant material in bread and carrots have been found within a few papers reporting on the use of natural mold in the products [47,48].

The aim of our study was to examine the antioxidant, antimicrobial, and antibiofilm activity of bitter orange essential oil (*Citrus aurantium*) against biofilm-forming bacteria *Stenotrophomonas maltophilia* and *Bacillus subtilis* by evaluating different biofilm-forming stages using MALDI-TOF MS, as well as *Penicillium* spp. Growth control by contact and the vapor method.

## 2. Results and Discussion

### 2.1. Chemical Composition of Bitter Orange Essential Oil (Citrus aurantium L., CAEO)

The main volatile compounds of the bitter orange essential oil (*Citrus aurantium* L.) (CAEO) based on the reduced percentage were linalyl acetate (63.37%), α-terpineol (8.84%), geranyl acetate (6.02%) and neryl acetate (3.77%) (Table 1). While Bourgou et al. [49] reported that the main volatile components of CAEO were linalool, linalyl acetate, and alpha terpineol, Rahimi et al. [50] detected linalool and α-tepineol as well. Sarrou et al. [51] stated that limonene (94.67%), β-myrcene (2.00%), linalool (0.67%), β-pinene (0.62%) and α-pinene (0.53%) were the main terpenes of the *C. auratium* peel essential oil grown in Greece. Bourgou et al. [49] concluded that differences in the quality and quantity of volatile ether components might be caused by genetic, environmental, geographical, and seasonal factors. In the study of Zarrad et al. [52], limonene (87.523%), linalool (3.37%), and β-myrcene (1.63%) were the main compounds in *C. aurantium* fruits of Tunisian origin.

### 2.2. Antioxidant Activity of CAEO

The DPPH radical inhibition value of *C. aurantium* essential oil was 13.93%. Khettal et al. [53] found that the antioxidant activity of *C. aurantium* was 13.34%, while Bendaha et al. [54] reported activity of 15.33%. Choi et al. [55] investigated the radical scavenging activities of 34 citrus essential oils by the DPPH assay, where four essential oils of *Citrus aurantium* L. showed scavenging effects from 17.7% to 34.1%.

### 2.3. Antimicrobial Assay

The highest antibacterial activity of CAEO against *Stenotrophomonas maltophilia* was 17.67 ± 0.58 mm, less than those for chloramphenicol-33.33 ± 1.53 mm. Antimicrobial activity was furthermore observed against *Bacillus subtilis* (15.67 ± 1.53 mm) followed by *P. crustosum* (10.67 ± 1.15 mm), *P. expansum* (8.67 ± 0.58 mm) and *P. citrinum* (7.33 ± 1.53 mm). In another study, the EOs extracted from the peel of *C. aurantium* showed little antimicrobial activity against *B. cereus, S. flexneri, K. pneumoniae*, and *Colletotrichum capsici* [56]. Gram-positive bacteria were generally more sensitive to the activity of the CAEO [57]. Previous reports suggest that the antimicrobial activity of CAEO depends on specific microbial species; however, the activity is strain-dependent [58,59,60], which is in agreement with our results.

### 2.4. Antibiofilm Activity of CAEO

The antibiofilm activity (minimal biofilm inhibition concentration, MBIC) values of 50 and 90 were 8.47 μL/mL and. 9.21 μL/mL for *S. maltophilia*, and 8.56 μL/mL and 9.18 μL/mL for *B. subtilis*, respectively. According to Fathi et al. [61], MBICs for *E. coli* and *K. pneumoniae* to *C. aurantium* essential oil in the study were 100 mg/mL and 150 mg/mL, respectively. The antimicrobial activity of CAEO reported by Gniewosz et al. [62] was 1.25 mg/mL MIC and 5 mg/mL MBC for *S. aureus*, 2.5 mg/mL and 5 mg/mL in case of *Saccharomyces cerevisiae*, and 2.5 mg/mL and 5 mg/mL MBC with respect to *S. cerevisiae*.

### 2.5. Studies on Biofilm Development and Molecular Differences on Surfaces after Treatment with C. aurantium EOs

The developmental stage spectra of *S. maltofilia* biofilm during the experiment period are shown in Figure 1. The experimental groups were treated with CAEO. The arrangement of spectra in pairs was made by evaluating the growth on different surfaces. Spectra of planktonic cells were obtained from the culture medium.

The agreement of the control planktonic spectrum with the experimental spectrum obtained from wood was observed for three days from the start of the experiment. In contrast, the glass spectrum was significantly different when compared to the planktonic spectrum. Furthermore, on day 5 (Figure 1B), it was observed that two experimental groups were significantly different from the control planktonic cells. In the following days (7–14), differences in the experimental groups compared to the control planktonic spectrum were also noted (Figure 1C–F).

CAEO was added to media containing *B. subtilis* cell culture. Figure 2 displays the MALDI-TOF spectra at the different stages of the *B. subtilis* biofilm development from day 3 to 14 of the experiment. Each pair of spectra (A–F) represents the same day of the biofilm growth. Experimental biofilm samples were taken from glass or wood solid surfaces. Planktonic cells that served as a control sample were obtained from liquid media. The mass spectra of experimental and control samples at days 3–7 (Figure 2A–C) exhibited the similarities. Degradation of the experimental spectra in comparison to the control was detectable following 9–12 days (Figure 2D,E), after which CAEO started to inhibit the growth of *B. subtilis* biofilm. Spectra from day 14 (Figure 2F) following the CAEO addition showed a complete degradation of bacterial biofilms in the experimental samples.

Altogether, we observed three main clusters divided into five subclusters on the dendrogram of *S. maltophilia* from the experimental group treated with CAEOs (Figure 3). The highest similarity of SMEs was recorded for the experimental group on day 3 with planktonic cells. All control groups except for those from day 14 were included in the same cluster. In the second cluster, experimental groups from day 5 were included. Experimental groups of days 5 and 7 showed larger MSP distances than groups of other days. The differentiation of these experimental groups into a separate cluster was induced by a change in the structure of the biofilm, due to the CAOE.

The *B. subtilis* dendrogram consists of two main clusters with a high degree of similarity recorded primarily on days 3 to 7. The constructed dendrogram (Figure 4) shows the same manner of grouping as in the case of the mass spectra, the similarity of the experimental group up to 7 days with the shortest MSP distances. In other samples, relatively different spectra were detected. Planktonic spectrum and control groups up to day 9 were included in the same main cluster. The second cluster contains separate experimental groups for days 12 and 14 in a separate branch.

MALDI-TOF MS Biotyper has been proved to be an effective method for microbial identification and biofilm development evaluation [63]. MALDI-TOF was able to detect the surface characteristics on which bacteria were grown and differentiate *S. maltophilia* biofilm development on plastic and glass surfaces. The authors found that the bacteria matured more rapidly on plastic surfaces and reached the dispersed stage in a shorter time. *S. maltophilia* attached to polystyrene within 2 h of incubation, and the biofilm formation increased over time, reaching the maximum growth at 24 h [63]. A different capacity of biofilm-production was reported in 24 *S. maltophilia* pulsotypes of isolates depending on their genetic characteristics [35]. *Bacillus subtilis* is an established model system to investigate the molecular mechanisms of biofilm formation and development. *Bacilus subtilis* growth kinetics and morphological features were characterized for colony type biofilm formation previously [64]. The complexity of *B. subtillis* biofilm is characterized by a rapidly developing three-dimensional complex structure with the core size remains largely unchanged while the colony expansion is mostly attributed to the growth in the area of outer cell subpopulations [65]. Oliveira et al. [66] indicated that CAEOs were effective in controlling multi-species biofilms.

Kačániová et al. [67] evaluated the effect of coriander oil on *S. maltophillia* and *B. subtilis* plankton and biofilm cells with a similar MALDI-TOF MS staining method on biofilm development. The authors found that this method was effective in distinguishing the changes in the spectra of bacterial biofilm development under specific conditions. Coriander essential oil expressed the antibacterial activity against *B. subtilis*, *S. maltophilia* and *Penicillium expansum*. The strongest antimicrobial activity was found against *B. subtilis*, while the strongest antibiofilm activity was detected against *S. maltophilia*.

### 2.6. Water Activity and Moisture Content

The moisture content of bread was 41.467 ± 0.881%, and water activity was 0.945 ± 0.002. After baking, the bread quality depends on a_w_ and moisture content; these parameters are known to directly affect the shelf life of the product [68]. Bread a_w_ indicates the water availability for microbial growth and represents a substantial impact on the growth of microorganisms in foods [69,70], and a_w_ below 0.7 shows a preventive effect on the microbial spoilage [71]. The value for a_w_ in white bread within the range of 0.94 to 0.97 [72] indicates that the bread is susceptible to microbial spoilage, mainly attributed to the growth of various molds.

According to Lahlali et al. [73], bread is a food product with intermediate moisture, typically ranging from 35–42% [74], which corresponds to our results. The moisture content of 42% and 41.89% for white bread reported by Lee et al. [75] and Jaekel et al. [76] corresponds to our findings.

The most common fungi found in bakery products include *Rhizopus, Aspergillus, Penicillium, Monilia, Mucor*, and *Eurotium* [77]. *Penicillium expansum* is more resistant to harsh environmental conditions in comparison with other representatives of *Penicillium* species; therefore, this fungus was used as a model in our study [78].

### 2.7. In Situ Antifungal Analysis on Bread

Minimum inhibitory doses (MID_50_ and MID_90_) of CAEO against *Penicillium crustosum* on bread after 14 days were 98.71 and 123.39 (Figure 5), 136.52 and 188.40 against *P. citrinum* and 353.12 and 564.99 against *P. expansum*, respectively. Previously reported MID_50_ and MID_90_ for coriander EOs for growth inhibition of *Penicillium expansum* on bread after 14 days were 367.19 and 445.92, respectively [67]. The CAEO can be applied in the production and processing of different foods of plant and animal origin based materials, e.g., meat, fish, cheese, and cosmetic industry, for the production of different hygienic products and fragrances. Multilateral physiological effects are well-known and widely utilized in medicine [57].

*Fusarium oxysporum, F. solani, F. avenaceum, Botrytis cinerea*, and *Bipolaris sorokiniana* were treated with the CAEO, and the activity of CAEO was variable depending on the applied dose of the essential oil and the fungal species [79].

### 2.8. In Situ Antimicrobial Effect on Carrot

The antimicrobial study of CAEO was determined in an in situ study. A better antibacterial activity of CAEO was found against *B. subtilis* with a concentration of 62.5 (Figure 6E). Previously reported antimicrobial activity of CAEO against *B. subtilis* ranged from 5 to 10 mg/mL [80]. The antimicrobial activity against *S. maltophilia* was demonstrated by cinnamon, thyme, and clove essential oils [81].

The best antifungal activity of CAEO was found against *P. crustosum* (Figure 6A–D). The growth of *P. chrysogenum* was inhibited by the application of citrus plants essential oils, grapefruit essential oil being the most active. Antimicrobial effects of essential oils of sweet orange [82,83] and pummelo [84] against *P. chrysogenum* and *P. expansum* were described as highly effective. Furthermore, the antifungal activity of CAEOs has been described against *Penicillium* spp. and *P. verrucosum* [85].

## 3. Materials and Methods

### 3.1. Essential Oil

The tested essential oil (*Citrus aurantium*, CAEO, bitter orange) was purchased from the company Hanus, a.s. located in Nitra, Slovakia. The chemical analysis of the CAEO was done prior to the detection of the biological activity.

### 3.2. Chemical Composition of Essential Oil

Gas chromatographic-mass spectrometric analysis of the essential oil was performed as in a previous work of Kačániová et al. [67]. The results are expressed as mean values of three injections ± standard errors (SE).

### 3.3. Radical Scavenging Activity-DPPH Method

The radical scavenging activity of the essential oil was determined using the 2,2-diphenyl-1-picrylhydrazyl (DPPH) assay, according to Sánchés-Moreno et al. [86] with small modifications [64].

### 3.4. Microorganisms

The bacterial species of *Stenotrophomonas maltophilia* and *Bacillus subtilis* were obtained from the milk industry and identified with the MALDI-TOF MS Biotype and 16S rRNA sequencing. The bacteria strains were tested for the antimicrobial resistance, antimicrobial, antioxidant, and antibiofilm activity. The fungi *Penicillium expansum, P. crustosum*, and *P. citrinum* were isolated from grape samples and identified with the MALDI-TOF MS Biotyper and 16S rRNA sequencing.

### 3.5. Antimicrobial Activity

The antibacterial activity was examined by the agar disc diffusion method. *S. maltophilia* and *B. subtilis* were incubated in Mueller Hinton broth (MHB, Oxoid, Basingstoke, UK) at 37 °C for 24 h. *P. expansum, P. crustosum*, and *P. citrinum* were incubated in Sabouraud broth (SB, Oxoid, Basingstoke, UK) at 25 °C for 48 h. The isolates were identified by useof the MALDI-TOF MS Biotyper (Bruker Daltonic, Bremen, Germany) with a score higher than 2.2. Microbial suspensions were prepared in saline and adjusted to 0.5 McFarland turbidity standards with a densilameter (Erba Lachema s.r.o., Brno, Czech Republic). The isolates were used for inoculation onto the Mueller–Hinton agar (MHA) and Sabouraud agars, subsequently discs impregnated by the tested EOs (10 µL/disc) were used for the detection of the antimicrobial activity. MHA were incubated at 4 °C for 1–2 h, and then at 37 °C and 25 °C for 18–24 h and 48 h for bacteria and microscopic fungi, respectively. The antimicrobial activity was evaluated by measuring the zone of growth inhibition around the discs following incubation. Chloramphenicol (30 µg, Oxoid, Basingstoke, UK) and fluconazole (25 µg, Oxoid, Basingstoke, UK) served as positive antimicrobial controls.

### 3.6. Minimum Biofilm Inhibitory Concentration (MBIC)

MBIC analysis was done on a microtitration plate. Taking advantage of the agar microdilution method, we determined the anti-biofilm activity of the CAEO for *S. maltophilia* [87]. The bacterial culture was cultivated on MHB at 37 °C for 24 h. After cultivation, bacterial suspension with a density of 10^8^ CFU/mL was prepared. One hundred μL of the bacterial suspension was transferred into a 96-well microtitration plate, subsequently, 100 µL of the essential oil at a concentration from 0.3125 μL to 10 μL per well were added. MHB with the essential oil was used as a negative control. As a control of the maximum growth, MHBs inoculated with microorganisms were used. The microplates were incubated at 37 °C for 24 h, and then the supernatant was poured away. The wells were washed 3 times with 250 µl of saline solution and dried for 30 min. Crystal violet (200 µL of 0.1% (*w*/*v*)) was subsequently added to each well of the dried microplate and allowed to stain for 15 min with a repeated wash. The samples were resolubilized with 200 μL 33% acetic acid [88]. The absorbance was measured with a spectrophotometer (Promega Inc., Madison, WI, USA) at 570 nm. The concentration of the essential oil at which the absorbance was less than or equal to the negative control was determined as MBIC. The test was performed in triplicate.

### 3.7. Biofilm Development and Molecular Differences on Different Surfaces with MALDI-TOF MS Biotyper

*S. maltophilia* and *B. subtilis* cultures were pre-inoculated in MHB at 37 °C for 24 h. Ten μL of each inoculum was added to 20 μL of MHB in 50 μL polypropylene tubes that contained a glass slide, and a wooden toothpick. 0.1% CAEO was added to the bacterial cultures. The prepared samples were incubated on a shaker at 170 rpm and 37 °C with a 45° inclination. Experimental samples of the biofilm and control specimens of planktonic cells were taken after 3, 5, 7, 9, 12, and 14 days.

Biofilm samples were wiped by a sterile cotton swab from a glass slide or wooden toothpick and subsequently applied in duplicate directly on a polished 96-well MALDI-TOF MS Biotyper plate (Bruker Daltonics, Bremen, Germany). Planktonic cells were obtained from planktonic cell suspension and centrifuged at 3000× *g* for 3 min. The pellet was used for the analysis with MALDI-TOF MicroFlex (Bruker Daltonics, Germany). A linear and positive mode for the range of *m/z* 200–2000 was used.

The standard global spectrum (MSP) of the samples was represented by 40 mass spectra generated for every sample by the MALDI Biotyper 3.0 software (Bruker Daltonics, Germany), the spectra were obtained automatically and analyzed with the MALDI Biotyper dendrogram method using Euclidean distances [63]. The measured spectra were compared to FlexAnalysis 3.0 (Bruker Daltonics, Germany).

### 3.8. Bread Making Process

The baking formula consisted of wheat flour T650 (250 g), water (150 mL), saccharose (2.5 g), salt (5 g), and yeast (5 g). All the ingredients were mixed in a spiralmixer (Diosna SP 12 D; Diosna; Germany) for 6 min. The prepared dough was placed into an aluminum vessel and transferred to a fermentation cabinet (MIWE cube; Pekass s.r.o., Plzeň, Czech Republic) at 32 °C and 85% relative humidity for 40 min. The loaves were baked in two stages: (i) At 180 °C for 17 min with the addition of 160 mL water and (ii) at 210 °C for 10 min in a steamy oven (MIWE cube). Following baking, the bread was left to stand at laboratory temperature for 2 h, and then analyzed.

### 3.9. Water Activity and Moisture Content

Cooled breadcrumbs water activity (a_w_) was measured with a Lab Master aw Standard (Novasina AG; Lachen, Switzerland) at 25 ± 0.3 °C. The moisture content was determined by the Kern DBS 60-3 moisture analyzer (Kern and Sohn GmbH, Balingen, Germany) at 120 °C.

### 3.10. In Situ Antifungal Analysis on Bread

The bread samples were cut into slices with a height of 150 mm and placed into 0.5 L sterile glass jars (Bormioli Rocco, Fidenza, Italy). A fungal spore suspension of each strain (final concentration of 1 × 10^6^ spores/mL) was diluted in 20 mL of sterile phosphate-buffered saline with 0.5% Tween 80 by adjusting the density to 1–1.2 McFarland and 5 µL of inoculum was used for bread inoculation. CAEO concentrations of 125, 250, and 500 µL/L (EOs + ethyl acetate) were evenly distributed on a sterile paper–filter disc (6 cm), which was inserted into the cover of the jar, except for the treatment of the control group. The jars were hermetically closed and kept at 25 °C ± 1 °C for 14 days in the dark. The colonies with visible mycelial growth and visible sporulation were evaluated [66].

### 3.11. Vapor Phase of Antimicrobial Assay with Carrot

Carrot samples were cut into slices of 5 mm. Warm PDA (potato dextrose agar; Oxoid, Basingstoke, UK) was poured into 60 mm Petri dishes (PD) and the lid, and 5 µL of inoculum prepared as described in chapter 2.10 were applied to the carrot slices. The EOs were two-fold diluted in ethyl acetate to a final volume of 250, 125, and 62.5 μL and transferred on a 55 mm round sterile filter paper using a micropipette. The filter paper was dried for 1 min for the evaporation of ethyl acetate and subsequently laid into the PD on the walls to ensure a 2 mm distance between the paper and the agar. Finally, the PD was hermetically closed with its lid containing the solidified agar. The Petri dishes were incubated at 37 °C for 18–24 h and at 25 °C for 72 h for bacteria and fungi, respectively.

### 3.12. Statistical Analysis

The measurements were carried out in triplicate. Statistical variability of the data was processed with Microsoft™ Excel^®^ program. MBIC50 and MBIC90 values (concentration causing 50% and 90% reduction of bacterial biofilm growth) were estimated by the logit analysis.

## 4. Conclusions

CAEO showed a satisfactory biological activity and inhibitory effects on biofilm formation. The results were confirmed by the protein spectrum analysis, cluster similarity analysis (dendrograms), and the antibiofilm activity of the essential oil. It can be assumed that the effect of CAEO is based on disrupting the polysaccharides in biofilms, thereby decreasing their resistance. MALDI-TOF MS profiling is a very convenient equipment to identify microorganisms and it is also suitable to analyze the biofilm profile.

MALDI-TOF MS Biotyper is a useful method to distinguish between different phases of the biofilm and to detect their transition to the dispersed stage. The present study highlights the possible use of MALDI-TOF technology in clinical diagnostics and prognostic processing of biofilm formation and/or control. The global concerns on antimicrobial resistance and health-related consequences facilitate a necessity to look for natural alternatives with antimicrobial properties. Our study has demonstrated the antimicrobial effect of CAEO that could be applicable for the food industry and health care needs.

## Figures and Tables

**Figure 1 molecules-25-03956-f001:**
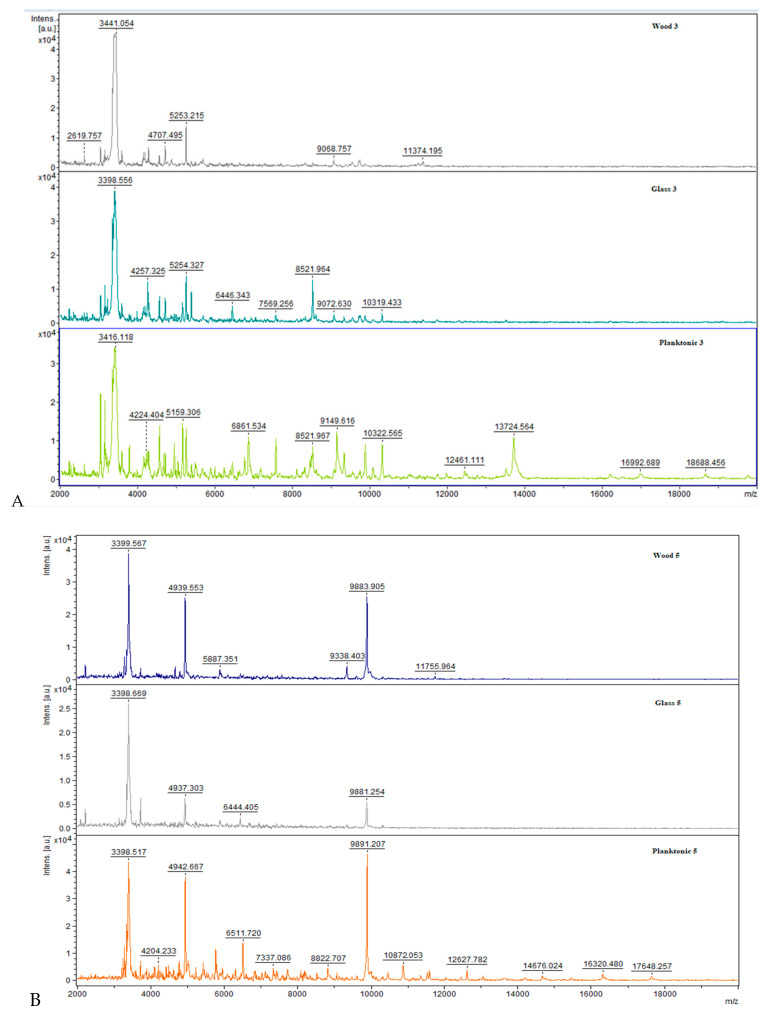
Representative MALDI-TOF mass spectra of *S. maltophilia*: (**A**) 3 day, (**B**) 5 day, (**C**) 7 day, (**D**) 9 day, (**E**) 12 day, (**F**) 14 day.

**Figure 2 molecules-25-03956-f002:**
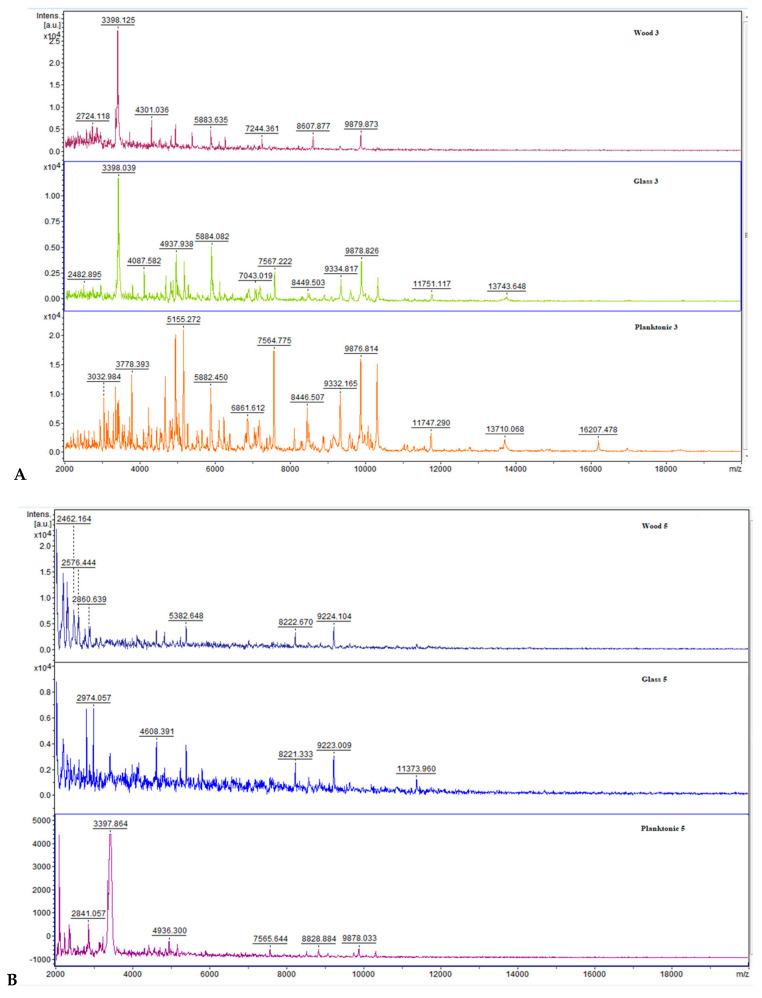
Representative MALDI-TOF mass spectra of *B. subtilis*: (**A**) 3 day, (**B**) 5 day, (**C**) 7 day, (**D**) 9 day, (**E**) 12 day, (**F**) 14 day.

**Figure 3 molecules-25-03956-f003:**
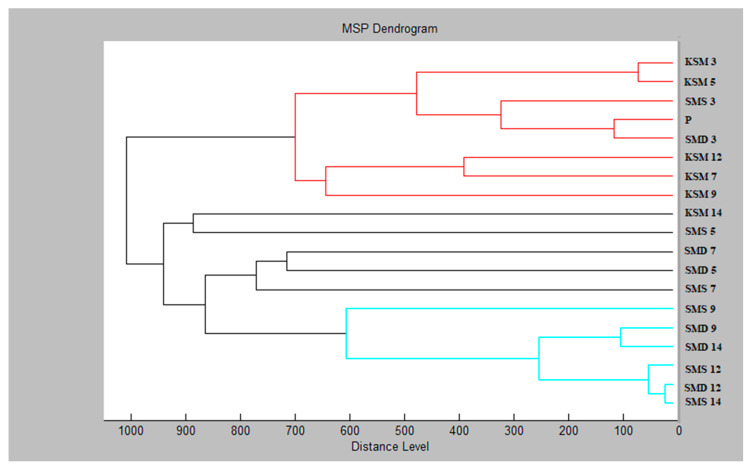
Dendrogram of *S. maltophilia* with MSP for planktonic cells and all experimental groups. Sample name abbreviations: K-control; SM-*Stenotrophomonas maltophilia*; S-glass; D-wood; P-planktonic cells.

**Figure 4 molecules-25-03956-f004:**
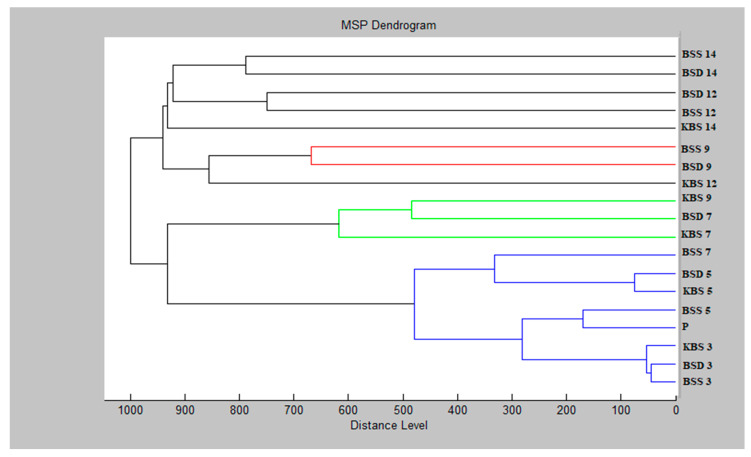
Dendrogram of *B. subtilis* with MSPs for planktonic cells and control. The sample names are abbreviated as follows: B, *B. subtilis*; K, control; S, glass; D, wood; P, planktonic cells.

**Figure 5 molecules-25-03956-f005:**
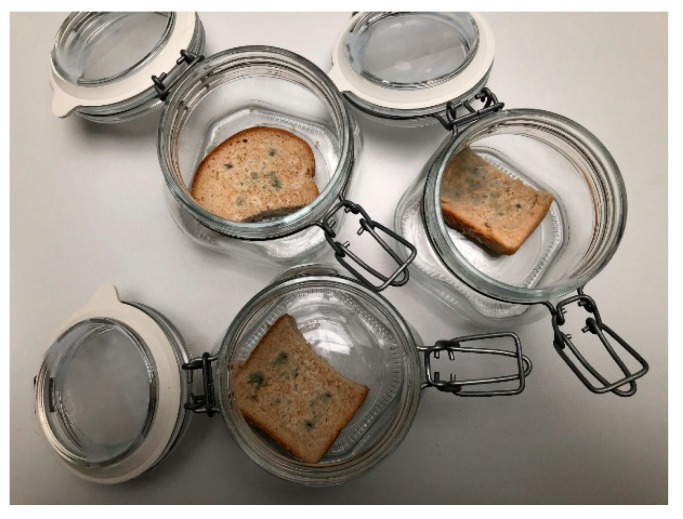
In situ antifungal analysis of bread inoculated with *Penicillium crustosum* in vapor phase.

**Figure 6 molecules-25-03956-f006:**
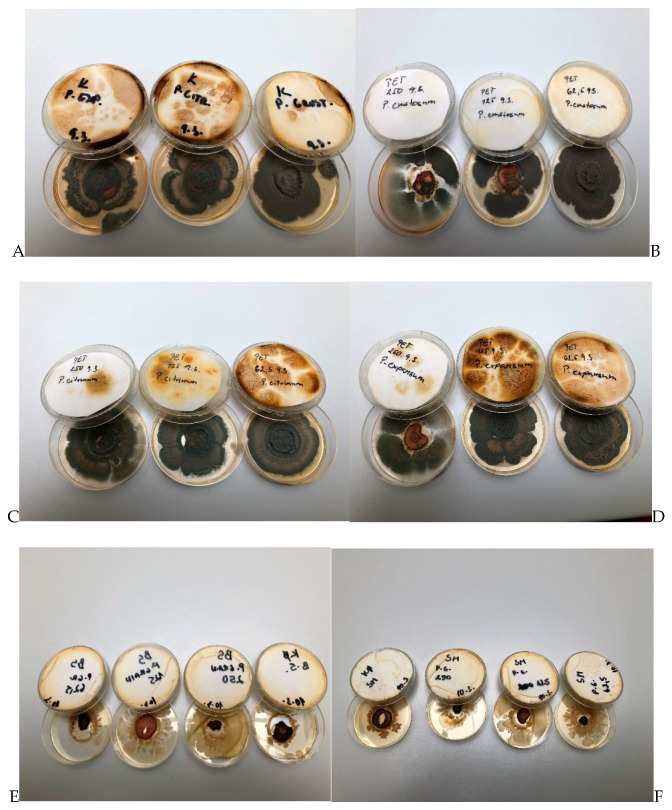
In situ antimicrobial analyses of carrot with *Penicillum* spp. in vapor phase ((**A**) control with fungi, (**B**) *P. crustosum*, (**C**) *P. citrinum*, (**D**) *P. expansum*, (**E**) *B. subtillis*, (**F**) *S. maltiphilia*).

**Table 1 molecules-25-03956-t001:** Main components of bitter orange essential oil (*Citrus aurantium*).

Name	Synonyms	TIC% Area ^a^
sabinene	4(10)-thujene	0.32 ± 0.06
3-carene		0.47 ± 0.09
β-myrcene		2.32 ± 0.31
d-limonene		1.57 ± 0.11
1,8-cineol	eucalyptol	2.70 ± 0.28
*cis*-ocimene		0.88 ± 0.07
β-*cis*-ocimene	ocimene-X	2.39 ± 0.21
α-terpinolene		0.58 ± 0.09
linalyl acetate		63.37 ± 2.78
caryophyllene		1.34 ± 0.13
(−)-α-terpineol	*p*-menth-1-en-8-ol	8.84 ± 0.67
bicyclogermacrene	isobicyclogermacrene, lepidozene, isolepidozene	0.33 ± 0.06
neryl acetate		3.77 ± 0.44
geranyl acetate		6.02 ± 0.52
*cis*-geraniol	Nerol neryl alcohol	1.63 ± 0.07
geraniol	*trans*-geraniol lemonol geranyl alcohol	3.68 ± 0.41

^a^ mean value ± SE.

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
