# Peer review of "Biological Activity and Antibiofilm Molecular Profile of Citrus aurantium Essential Oil and Its Application in a Food Model"

_molecules, 2020, doi:10.3390/molecules25173956_

Round 1
Reviewer 1 Report
The present manuscript deals with the chemical composition, antioxidant and antibiofilm activity of bitter orange Citrus aurantium essential oil and its application to food model. The results are of interest, however there are some of points which need additional attention by the Authors.
- In the abstract, the first sentence must be re-written. If the title is about C. aurantium, it is strange to mention the investigation “of biofilm profile of Stenotrophonomonas maltophilia and Bacillus subtilis” as first aim of the study.
- In the Abstract, the authors claim to have studied antibacterial activity by disc diffusion method and micordilution method. As a matter of fact, they used microdilution to study antibiofilm activity.
- The Authors should explain the choice of B. subtilis as test object.
- In Table 1, it is not clear how many replicates of the analysis were performed. Standard deviations should be presented.
Author Response
The present manuscript deals with the chemical composition, antioxidant and antibiofilm activity of bitter orange Citrus aurantium essential oil and its application to food model. The results are of interest, however there are some of points which need additional attention by the Authors.
Point 1: In the abstract, the first sentence must be re-written. If the title is about C. aurantium, it is strange to mention the investigation “of biofilm profile of Stenotrophonomonas maltophilia and Bacillus subtilis” as first aim of the study.
Response: The aim of study was specified as “The main aim of the study was to investigate the chemical composition, antioxidant, antimicrobial and antibiofilm activity of Citrus aurantium essential oil (CAEO).”
Point 2: In the Abstract, the authors claim to have studied antibacterial activity by disc diffusion method and microdilution method. As a matter of fact, they used microdilution to study antibiofilm activity.
Response: Abstract was corrected by adding “The antimicrobial activity was analysed by disc diffusion and the antibiofilm activity was assessed using microdilution methods, and the vapor phase was applied for three penicillia”
Point 3: The Authors should explain the choice of B. subtilis as test object.
Response: B. subtilis is a spore-forming bacterium, widespread in the environment and well-known because of its ability to form biofilms. The main area of interest of authors was to compare spore-forming and non-spore-forming biofilm-producing isolates.
Point 4:
In Table 1, it is not clear how many replicates of the analysis were performed. Standard deviations should be presented.
Response: The results are expressed as mean values of three injections ± standard errors (SE). Corrections were made in “Materials and methods” section as well as in Table 1.
Reviewer 2 Report
In my opinion, the present paper is very similar to the previous work of the authors https://www.mdpi.com/2304-8158/9/3/282.
In my opinion, this paper has two much matches (34%) with other papers (please, see red text in the attached file). Maybe, it is a self-plagiarism, but I think it should not be so.

Author Response
Point 1: In my opinion, the present paper is very similar to the previous work of the authors https://www.mdpi.com/2304-8158/9/3/282.
In my opinion, this paper has two much matches (34%) with other papers (please, see red text in the attached file). Maybe, it is a self-plagiarism, but I think it should not be so.
Response: Our group has been working intensively with antimicrobial activity of essential oils, therefore some part of the manuscript, especially of the “Material and methods” section, might contain self-plagiarism, as the applied methods are similar for examination of properties of essential oils. The manuscript was carefully revised with suspicious parts of the manuscript were shorten or revised.
Reviewer 3 Report
After evaluating the manuscript, I consider that the results are preliminary. The mass spectra data obtained by MALDI-TOF need to be evaluated to understand chemically the effects of the essential oil of C. aurantium on the chemical profiles of the two microorganisms. Total ion chromatograms must be generated by Auto MS and MS/MS experiments, later converted to MzXML and inserted in the GNPS platform. The results will allow monitoring the production of compounds from the experiments and identifying them. Subsequently, those considered as new for the respective microorganisms must be isolated for structural confirmation. It is also necessary to use liquid chromatography-atmospheric pressure chemical ionization-selected reaction monitoring (LC-APCI-MS-SRM) spectrometric to locate possible compounds from the C. aurantium essential oil in the microorganism chemical profile. After all these studies, the authors will have conclusive data to understand the interaction between the C. aurantium essential oil and the microorganisms studied, and how the constituents of the oil affect their chemical profile during the biofilm formation for several days.
I suggest the authors consult the reference below.
WANG, M.; CARVER, J. J.; PHELAN, V. V; et al. Sharing and community curation of mass spectrometry data with Global Natural Products Social Molecular Networking. Nature Biotechnology, v. 34, n. 8, p. 828–837, 2016.
Author Response
Point 1: After evaluating the manuscript, I consider that the results are preliminary. The mass spectra data obtained by MALDI-TOF need to be evaluated to understand chemically the effects of the essential oil of C. aurantium on the chemical profiles of the two microorganisms. Total ion chromatograms must be generated by Auto MS and MS/MS experiments, later converted to MzXML and inserted in the GNPS platform. The results will allow monitoring the production of compounds from the experiments and identifying them. Subsequently, those considered as new for the respective microorganisms must be isolated for structural confirmation. It is also necessary to use liquid chromatography-atmospheric pressure chemical ionization-selected reaction monitoring (LC-APCI-MS-SRM) spectrometric to locate possible compounds from the C. aurantium essential oil in the microorganism chemical profile. After all these studies, the authors will have conclusive data to understand the interaction between the C. aurantium essential oil and the microorganisms studied, and how the constituents of the oil affect their chemical profile during the biofilm formation for several days.
I suggest the authors consult the reference below.
WANG, M.; CARVER, J. J.; PHELAN, V. V; et al. Sharing and community curation of mass spectrometry data with Global Natural Products Social Molecular Networking. Nature Biotechnology, v. 34, n. 8, p. 828–837, 2016.
Response: Our further research will be focused on metabolomic studies related to chemical profiles of two tested microorganisms, as well as possible effects of some constituents of essential oils on the biofilm confirmation. For this purpose we will use MALDI TOF (Auto MS and MS/MS experiments) and nanoLC HRMS, LC HRMS (in this case, it will include the APCI technique of ionization) for identification of selected terpenoid compounds from essential oil on the chemical profile of tested microorganisms. Thank you very much for such a favorable review. The purpose of this research is not a mass spectrometry-associated metabolic study of treated bacteria. In the future for serious research MALDI-TOF MS Biotyper may give lot of possibilities and we can think about this.
Reviewer 4 Report
Dear Authors,
my comments are in the attached file.
Best regards

Author Response
The study addresses an interesting topic which falls within the scope of “Molecules” journal. Authors’ objective is to characterise Citrus aurantium Essential Oil and to show its biological and amimicrobial activities.
Point 1: The authors have highlighted the aims, significance and the novelty of their work. The conclusions made are supported by the data presented, however, there are several issues need to be revised and points to be detailed.
Major points for discussion
Point 2: 1. Abstract structure should be rephrased. Authors mentioned CAEO investigation against microscopic fungi only at the end and not at the beginning in the aims
Response: The aim of the study was rewritten and CAEO was added.
Point 3: 2. Introduction should be improved. Authors never mentioned penicillum spp (P. crustosum, expansum and citrinum) characteristics as instead done for S. matophila and B subtilis. Little space has been reserved for the fungi Minor points for discussion
Response: Lines 83-86 were added to highlight the topicality of Penicillium spp. research.
Point 4: 1. Improvement of the English language and scientific mother tongue editing are necessary
- Line 199. The word “approach” should be removed
- Line 299. The word “tested” should be removed
- Line 301. “16S rRNA sequencing” should replace “16S sequencing”; to this regard, which variable regions have been sequenced?
- Line 303. “The fungus Penicillium expansum, P. crustosum and P. citrinum were ..”. Fungi and not fungus is correct
- Line 380. “Bormioli” is the correct name instead of “Bromioli”
- Line 390. “Carrot samples were cut into slices of 50 mm”. Is this dimension correct?
- Line 393. “EOs were diluted in ethyl acetate using a two-fold dilution manner to give the final volume of 250, 125 and 62.5 μL” should be rephrased into ““EOs were two-fold diluted in ethyl acetate to give final volumes of 250, 125 and 62.5 μL”
- Line 404. What are the p values considered as significant?
Response: All comments were mentioned in point 4 were introduced into the manuscript.
Round 2
Reviewer 2 Report
Latin names must be in italic in the reference list.
Author Response
Point 1: Latin names must be in italic in the reference list.
Response: It has been corrected.
Reviewer 3 Report
The authors' response is not convincing. I still consider that the manuscript contains only preliminary data.
The mass spectra data obtained by MALDI-TOF need to be evaluated to understand chemically the effects of the essential oil of C. aurantium on the chemical profiles of the two microorganisms. Total ion chromatograms must be generated by Auto MS and MS/MS experiments, later converted to MzXML and inserted in the GNPS platform. The results will allow monitoring the production of compounds from the experiments and identifying them. Subsequently, those considered as new for the respective microorganisms must be isolated for structural confirmation. It is also necessary to use liquid chromatography-atmospheric pressure chemical ionization-selected reaction monitoring (LC-APCI-MS-SRM) spectrometric to locate possible compounds from the C. aurantium essential oil in the microorganism chemical profile. After all these studies, the authors will have conclusive data to understand the interaction between the C. aurantium essential oil and the microorganisms studied, and how the constituents of the oil affect their chemical profile during the biofilm formation for several days.
Author Response
Point 1: The authors' response is not convincing. I still consider that the manuscript contains only preliminary data. The mass spectra data obtained by MALDI-TOF need to be evaluated to understand chemically the effects of the essential oil of C. aurantium on the chemical profiles of the two microorganisms. Total ion chromatograms must be generated by Auto MS and MS/MS experiments, later converted to MzXML and inserted in the GNPS platform. The results will allow monitoring the production of compounds from the experiments and identifying them. Subsequently, those considered as new for the respective microorganisms must be isolated for structural confirmation. It is also necessary to use liquid chromatography-atmospheric pressure chemical ionization-selected reaction monitoring (LC-APCI-MS-SRM) spectrometric to locate possible compounds from the C. aurantium essential oil in the microorganism chemical profile. After all these studies, the authors will have conclusive data to understand the interaction between the C. aurantium essential oil and the microorganisms studied, and how the constituents of the oil affect their chemical profile during the biofilm formation for several days.
Response: The aim of the present study was to study chemical composition of essential oil for detection of antimicrobial and antibiofilm activity on microorganisms. Chemical composition of essential oils has being studied with gas chromatographic-mass spectrometric analysis. MALDI-TOF MS Biotyper was used for evaluation of proteins spectra of microorganisms and biofilms of interest. MALDI TOF MS was applied for identification of microorganisms and their products and not chemical analysis.
Our further research will be focused on metabolomic studies related to chemical profiles of two tested microorganisms, as well as possible effects of some constituents of essential oils on the biofilm formation. For this purpose, we will use MALDI TOF (Auto MS and MS/MS experiments) and nanoLC HRMS, LC HRMS (in this case, it will include the APCI technique of ionization) for identification of selected terpenoid compounds from essential oil on the chemical profile of tested microorganisms.